# Frequency of depressive symptoms in Syrian refugees and Turkish maintenance hemodialysis patients during COVID-19 pandemic

Mustafa Sevinc[1], Nuri Baris Hasbal[2], Tamer Sakaci[1]*, Taner Basturk[3], Elbis Ahbap[1], Mustafa Ortaboz[1], Emrah Erkan Mazi[1], Efruz Pirdogan[4], Jonathan Ling[5], Abdulkadir Unsal[3]

1 Nephrology Department, Sisli Hamidiye Etfal Education and Training Hospital, Istanbul, Turkey,
2 Nephrology Department, Basaksehir Cam and Sakura City Hospital, Istanbul, Turkey, 3 Nephrology Department, University of Health Sciences, Sisli Hamidiye Etfal Education and Training Hospital, Istanbul, Turkey, 4 Psychiatry Department, Sisli Hamidiye Etfal Education and Training Hospital, Istanbul, Turkey, 5 Faculty of Mental Health and Wellbeing, University of Sunderland, Sunderland, United Kingdom

* sakacitamer@hotmail.com

## Abstract

## Introduction

Pneumonia of unknown cause was detected on 30 December 2019 in China. It was categorized as an outbreak and named as COVID-19 by the World Health Organization. The pandemic affects all people, but patient groups such as hemodialysis (HD) patients have been particularly affected. We do not know if refugees suffered more during the outbreak. In this study, we compared depressive symptom frequency between Syrian refugee HD patients and Turkish ones.

## Methods

The study had a single-center, cross-sectional design. Demographic and clinical data were collected retrospectively from patients' files containing details about past medical history, demographic variables and laboratory values. Validated Turkish and Arabic forms of Beck Depression Inventory (BDI) were used to assess depressive symptoms. BDI scores were compared according to nationality, demographic features and clinical data. A BDI score more than 14 was accepted as suspicion of depression.

## Results

119 patients were enrolled in the study. After the exclusion of 22 patients, 75 Turkish and 22 Syrian patients were included for further analysis. The median BDI (interquartile range) score for Turkish and Syrian patients were 12 (7–23) and 19.5 (12.7–25.2), respectively (p = 0.03). Suspicion of depression was present at 42.7% of Turkish, and 72.7% of Syrian HD patients (p = 0.013). Regarding all patients, phosphorus level, Kt/V, and nationality were

**Data Availability Statement:** We have uploaded our date via Dryad system by doi:10.5061/dryad. wwpzgmshv.

**Funding:** The author(s) received no specific funding for this work.

**Competing interests:** The authors have declared that no competing interests exist.

significantly different between patients with and without suspicion of depression (p = 0.023, 0.039, 0.013, respectively).

## Conclusion

Syrian patients had higher BDI scores and more depressive symptoms than Turkish patients. Additional national measures for better integration and more mental support to Syrian HD patients are needed.

## Introduction

Pneumonia of unknown cause was detected on 30 December 2019 in China. It was categorized as an outbreak and named as COVID-19 by the World Health Organization [1]. The first COVID-19 case was reported on 10 March 2020 in Turkey [2]. People older than 65 years old began their lockdown on 21 March 2020 and people younger than 20 years old started to lock-down on 03 April 2020. Total lockdown has been done a few times lasting for three or four days, especially at weekends.

Renal replacement modalities have been affected in different ways during the pandemic. Patients with kidney transplantation, performing home hemodialysis (HD) and peritoneal dialysis have been advised to isolate themselves at homes. However, center HD patients continued to attend dialysis centers.

Depression is a frequent co-morbidity at HD patients. It can be screened by a few types of self-questionnaires validated in this population such as Beck depression inventory (BDI) [3]. The frequency of depressive symptoms during pandemic on chronic HD patients is not known.

The Syrian civil war started on 11 March 2011. Turkey currently hosts 3.6 million registered Syrian refugees [4]. There are many Syrian center HD patients both in Turkey and in our unit. The number of Syrian refugees under HD therapy was 345 in 2019 [5]. We do not know if there is a difference in depressive symptom frequency between Syrian refugee HD patients and Turkish ones or not. This study aims to compare BDI scores of Turkish and Syrian mainte-nance HD patients during the COVID-19 pandemic.

## Methods

This cross-sectional study (Clinicaltrials.gov: NCT04444557) was approved by the local ethics committee of Sisli Hamidiye Etfal Education and Research Hospital on 12 May 2020 (Ref. number 1514). All data were fully anonymized before access. Participants signed written consent forms both to participate in the study and to have data from their medical records used in research.

All HD patients in the same center were invited to participate in the study. Exclusion crite-ria were as follows: patients younger than 18 years old, patients undergoing home-HD, history of HD less than three months, inability to complete the questionnaire, nationalities other than Turkish and Syrian, not volunteering to fill the form, history of hospitalization due to any rea-son during pandemic time starting from 10 March 2020, and history of COVID-19 before questionnaire. BDI forms were filled between 17 April 2020 and 12 May 2020. Past medical records were accessed at the same day patients completed the BDI questionnaire.

Demographic data were collected retrospectively from patients' files. These files contain the demographic details of patients filled at admission to our dialysis center and patients' past and

current laboratory values. Age, sex, nationality, marital status, education level, height, weight, etiology of kidney disease, the date for initiation of center HD, last hemoglobin, albumin, phosphorus, parathyroid hormone, Kt/v, creatinine levels and, Charlson comorbidity index were noted. Marital status was grouped as married and not married including single, divorced, and widow.

BDI consists of 21 questions and every question has four choices ranging from 0 to 3 in which 0 represents the absence of a problem, and 3 represents an extreme problem. Therefore, the total BDI score can be between 0 and 63 points. BDI score of more than 14 was accepted as suspicion of depression in patients with end-stage renal disease (ESRD) [6–8]. The frequencies of depressive symptoms were compared between nationalities. The validated Turkish and Arabic forms of BDI version-I were filled by patients themselves [9–11]. If help was required to complete the questionnaire, a native Arabic physician working at our hemodialysis center (SS) and the official Arabic translator of our institution (SA) helped Syrian patients.

BDI score was compared according to age group (<65 years or ≥65 years), sex, education level, marital status, body mass index (<25 kg/m$^2$, 25–30 kg/m$^2$, >30 kg/m$^2$), presence of diabetes mellitus, presence of hypertension, HD vintage (below or above median), hemoglobin level (<10 g/dl, 10–12 g/dl, >12 g/dl), phosphorus level (<3.5 mg/dl, 3.5–5.5 mg/dl, >5.5 mg/dl), albumin level (<3.5 gr/dl or ≥3.5 gr/dl), parathyroid hormone level (below or above median), Kt/v (<1.4 or ≥1.4), and nationality. BDI score was also divided into cognitive-affective and somatic-performance subscales [12]. These were compared to nationality as well.

## Statistical analysis

Statistical analyses were performed with the Scientific Package for Social Science (version 21.0; SPSS Inc., Chicago, IL, USA). Continuous variables were given as mean ± standard deviation if they were distributed normally or as median (interquartile range) if they were distributed abnormally. Qualitative variables were given as a percentage. A comparison of normally distributed data was performed by independent samples *t*-test. Abnormally distributed data was compared with the Mann-Whitney *U* test. Categorical variables were compared by the Chi-Square test. Differences were considered statistically significant for *p* values less than 0.05.

## Results

One hundred and nineteen patients were enrolled to the study. After the exclusion of 22 patients for a range of reasons, 75 Turkish and 22 Syrian patients were included for further analysis (Fig 1). The mean age of patients was 51.6 ± 15.5 years. The most common cause of ESRD was hypertension (32%). Median HD vintage was 3.7 (1.8–7.4) years (Table 1).

The mean age for Turkish patients was 52.1 ± 16.1 years and 49.9 ± 13.2 years for Syrian patients (p = 0.568) (Table 1). Female patients were 52.0% of the Turkish cohort and 18.2% of the Syrian cohort (p = 0.005). The education level was similar between the two nationalities (p = 0.06). Marriage rate was 49.3% in Turkish and 81.8% in Syrian patients (p = 0.007).

The median BDI score for Turkish patients was 12 (7–23). It was 19.5 (12.7–25.2) for Syrian patients (p = 0.03) (Fig 2). The median somatic-performance subscale scores for Turkish and Syrian patients were 2 (1–5) and 3.5 (2–6.2), respectively (p = 0.02). Affective-cognitive subscale median score for Turkish patients was 9 (6–18) whereas it was 14 (10–18) for Syrian patients (p = 0.05).

Depressive symptoms were present in 49.5% of the total cohort. Patients with and without depressive symptoms were compared on multiple variables (Table 2). Phosphorus level and Kt/V were found different between two groups (p = 0.023, 0.039, respectively). Depressive symptoms were present at 42.7% of Turkish, and 72.7% of Syrian HD patients (p = 0.013).

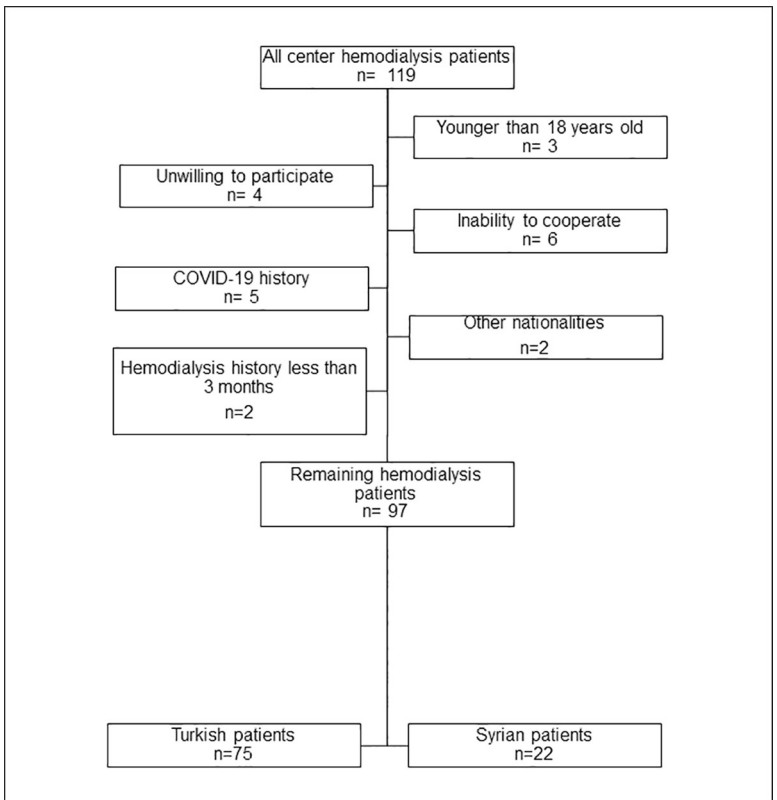

**Fig 1. Details of the patient cohort.** After enrollment of all center hemodialysis patients (n = 119), 22 patients were excluded due to following reasons: unwilling to participate (n = 4), younger than 18 years old (n = 3), inability to cooperate (n = 6), COVID-19 history (n = 5), hemodialysis history less than 3 months (n = 2), other nationalities (n = 2). Seventy-five Turkish, 22 Syrian patients' data were analyzed further.

## Discussion

This study investigated BDI scores between Turkish and Syrian HD patients during the COVID-19 pandemic. Syrian patients had higher BDI scores than Turkish patients.

After Turkey has started to accept refugees from Syria, free healthcare facilities including hemodialysis and kidney transplantation were established [13]. For HD, Syrian and Turkish patients have equal conditions including HD schedules (three times a week, four hours), and transport facilities. Drugs are supplied free of charge to Syrian patients, like Turkish ones. The groups were similar in many respects, with laboratory features hemoglobin, phosphorus, albumin, parathyroid hormone, Kt/V in both groups not differing significantly between the Syrian and Turkish patients. In our cohort, most of the Syrian patients (81.8%) were male, although 54.0% of all Syrian refugees [14] and 51.6% of Syrian hemodialysis patients [5] in Turkey were male. There could be a male predominance by chance in our dialysis center.

Screening frequency of depressive symptoms by the BDI score without a clinical interview is an established strategy in ESRD patients [3, 7, 8]. BDI-I questionnaire was used in this study because of some advantages. It was a practical, easy, reliable and valid self-report system that has been used since 1961 [11, 15]. It could be completed by patients themselves which would minimize close and long contacts with patients during the pandemic and prevent potential COVID-19 transmission. Moreover, to the best of our knowledge, it is the only self-report screening measure that has been validated in both Turkish and Arabic languages. Besides these advantages, BDI-I form has some limitations. It has been focused more on cognitive and

**Table 1. Demographic characteristics, laboratory values of all patients, Turkish, and Syrian subgroups.**

| Parameter | All patients (n = 97) | Turkish (n = 75) | Syrian (n = 22) | p |
|---|---|---|---|---|
| **Age, years, mean ± SD** | 51.6±15.5 | 52.1±16.1 | 49.9±13.2 | 0.568 |
| **Female patients (%)** | 44.3 | 52 | 18.2 | **0.005** |
| **Education level (%)** | | | | 0.060 |
| Illiterate | 6.0 | 4.8 | 9.5 | |
| Literate | 15.7 | 16.1 | 14.3 | |
| Primary school | 49.4 | 53.2 | 38.1 | |
| Secondary school | 13.3 | 14.5 | 9.5 | |
| High school | 12 | 11.3 | 14.3 | |
| University | 3.6 | - | 14.3 | |
| **Marital status** | | | | **0.007** |
| **Married (%)** | 56.7 | 49.3 | 81.8 | |
| **BMI, kg/m$^2$, median (IR)** | 21.8 (19.2–25.0) | 21.9 (19.3–25.3) | 21.1 (19.1–24.2) | 0.435 |
| **ESRD etiology (%)** | | | | 0.905 |
| Diabetes | 18.6 | 20 | 13.6 | |
| Hypertension | 32 | 32 | 31.8 | |
| Glomerulonephritis | 7.2 | 6.7 | 9.1 | |
| Polycystic kidney disease | 7.2 | 8 | 4.5 | |
| Unknown | 14.4 | 12 | 22.7 | |
| Miscellaneous | 20.6 | 21.3 | 18.2 | |
| **Diabetes mellitus (%)** | 21.6 | 24 | 13.6 | 0.299 |
| **Hypertension (%)** | 47.4 | 48 | 45.5 | 0.833 |
| **Hemodialysis vintage, years, median (IR)** | 3.7 (1.8–7.4) | 4.0 (1.9–10.4) | 2.7 (1.6–5.3) | 0.112 |
| **Hemoglobin, (g/dl), mean ± SD** | 10.7±1.6 | 10.6±1.6 | 10.8±1.7 | 0.680 |
| **Phosphorus, (mg/dl), mean ± SD** | 5.5±1.5 | 5.3±1.5 | 6.0±1.5 | 0.085 |
| **Albumin, g/dl, median (IR)** | 3.7 (3.5–3.9) | 3.7 (3.5–3.9) | 3.8 (3.5–4.1) | 0.152 |
| **Parathyroid hormone, (ng/L) median (IR)** | 537.5 (343.7–786.7) | 548 (346.7–796.5) | 484.5 (309.7–750.0) | 0.420 |
| **Kt/V, mean ± SD** | 1.7±.0.2 | 1.7±0.2 | 1.6±0.2 | 0.056 |
| **Creatinine, mg/dl, median (IR)** | 9.1 (6.8–10.4) | 9.0 (6.5–10.2) | 9.4 (7.4–10.9) | 0.180 |
| **Charlson comorbidity index, median (IR)** | 3 (2–5) | 4 (2–5) | 3 (2–4) | 0.171 |

BMI: body mass index

ESRD: end-stage renal disease

IR: interquartile range

SD: standard deviation

affective symptoms than somatic functions. Furthermore, it could be biased by organic symptoms. It was updated as BDI-II form at 1996 in response to American Psychiatric Association's publication of DSM-IV criteria for major depressive disorder [16]. Despite this updated version, BDI-I is still widely used in different patient populations [17].

Patients with and without depressive symptoms had similar demographic and laboratory features except for phosphorus level, Kt/V, and nationality. Some similar studies found no association between depressive symptoms and phosphorus level or Kt/V although the number of participants was less than our cohort and the definition of suspicion of depression was different from each other in all studies [18–21]. The difference between our study and previous studies may be due to these reasons. Furthermore, there are many additional probable reasons including adherence to diet and prescribed drugs, quality of hemodialysis, etc. These factors, unfortunately, were not in the scope of the trial design.

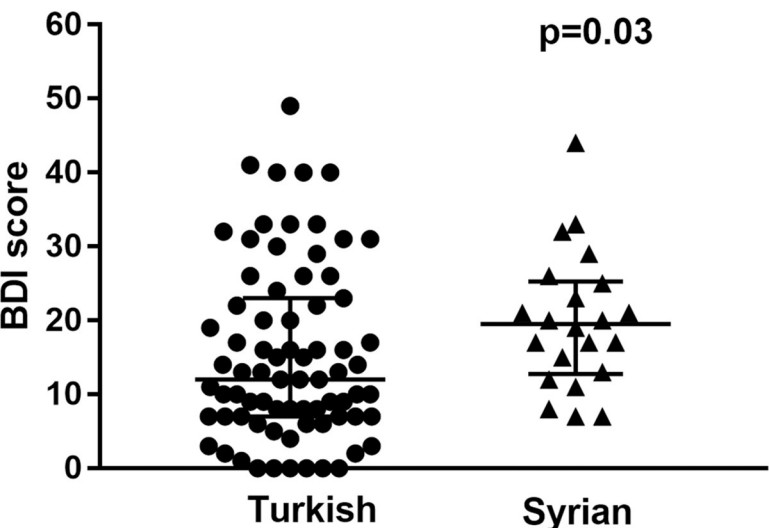

**Fig 2. Distribution of BDI scores for nationalities.** Every circle represents a Turkish, a triangle represents a Syrian patient. The median BDI score was 12 (7–23) for Turkish patients, 19.5 (12.7–25.2) for Syrian patients (p = 0.03).

Frequency of depressive symptoms in HD population is reported between 0 and 100% depending on the study and assessment tool [6]. Although it is unknown during the COVID-19 pandemic in the HD population, depressive symptom frequency was found between 18.7–35.4% in population-based studies [22–26]. Mazza et al have demonstrated that the risk of depressive symptoms was increased 22% more if medical problems were positive [27] which might be translated as depressive symptoms were expected more in HD population.

Mental health studies with the Syrian refugee population in Turkey out of the pandemic time were found depressive symptom frequency as 36.1%, 37.4%, and 70.5% [28–30]. There is a sensitivity for the protection of both physical and mental health of refugees or immigrants during the COVID-19 pandemic [31–34] although no published studies could be found.

Yilmaz et al have shown that Syrian maintenance HD patients were less compliant with their HD schedules [35]. Their per capita income was lower than Turkish patients, although their employment rate was higher. The number of Syrian household members was higher than Turkish patients but hot water accessibility was lower for the Syrian group. Similar BDI scores might be expected between Turkish and Syrian patients if the conditions were the same for both groups. However, providing equal physical and financial conditions only at peridialysis environment seemed to be insufficient for equal score expectation because we found that total BDI score, somatic-performance subscale score and affective-cognitive subscale score were all higher in Syrian patients than Turkish ones. As a result, it seems that we need to provide many other facilities to talk about equality between the two groups. These might be steps to improve language barrier between adult Syrian patients and Turkish service providers at all facilities including hospitals and other institutions. This could be with two methods: enabling Syrians to learn Turkish and maintaining more translation support at every institution or integrating native Syrian employees to these institutions like hospitals. Language support would help integration with the Turkish population. Most importantly, they should have been accepted and behaved like as others in Turkey, not as refugees. Integrating 4 million people into an economically hard stressed country in the midst of a pandemic is not easy but it seems as a necessity.

To the best of our knowledge, this is the first HD study comparing Turkish patients' and Syrian refugees' depressive symptoms. The main limitation of this study is the absence of control BDI scores of this cohort out of the pandemic period. Even though it will not change the

**Table 2. Demographic and laboratory characteristics of patients with and without depressive symptoms.**

| | BDI score ≤14 | BDI score >14 | p |
|---|---|---|---|
| **Age, years, mean ± SD** | 50.5±15.8 | 52.7±15.1 | 0.484 |
| **Female patients (%)** | 40.8 | 47.9 | 0.482 |
| **Education level (%)** | | | 0.154 |
| Illiterate | 6.8 | 5.1 | |
| Literate | 13.6 | 17.9 | |
| Primary school | 52.3 | 46.2 | |
| Secondary school | 9.1 | 17.9 | |
| High school | 18.2 | 5.1 | |
| University | - | 7.7 | |
| **Marital status** | 53.1 | 60.4 | 0.465 |
| **Married (%)** | | | |
| **BMI, kg/m$^2$, median (IR)** | 21.6 (19.0–26.1) | 21.8 (19.4–24.6) | 0.907 |
| **Diabetes mellitus (%)** | 18.4 | 25.0 | 0. 428 |
| **Hypertension (%)** | 49.0 | 45.8 | 0.561 |
| **Hemodialysis vintage, years, median (IR)** | 3.6 (1.9–7.2) | 3.9 (1.7–9.8) | 0.900 |
| **Hemoglobin, (g/dl), mean ± SD** | 10.8±1.6 | 10.5±1.6 | 0.500 |
| **Phosphorus, (mg/dl), mean ± SD** | 5.9±1.5 | 5.1±1.5 | **0.023** |
| **Albumin, g/dl, median (IR)** | 3.8 (3.6–4.0) | 3.7 (3.5–3.9) | 0.394 |
| **Parathyroid hormone, (ng/L) median (IR)** | 518 (368–799) | 546 (312–766) | 0.629 |
| **Kt/V, mean ± standard deviation** | 1.7±0.2 | 1.6±0.2 | **0.039** |
| **Creatinine, mg/dl, median (IR)** | 9.3 (7.1–10.5) | 8.7 (6.5–10.2) | 0.194 |
| **Charlson co morbidity index, median (IR)** | 3 (2–5) | 4 (2.2–5) | 0.120 |
| **Nationality** | | | **0.013** |
| Turkish | 57.3 | 42.7 | |
| Syrian | 27.3 | 72.7 | |

BDI: Beck Depression Inventory

BMI: body mass index

ESRD: end-stage renal disease

IR: interquartile range

SD: standard deviation

main result that the Syrian patients have had more depressive symptoms during the COVID-19 pandemic, we cannot infer how COVID-19 affected the results exactly. Control BDI scoring was not performed because the number of cases continues to be high enough to ignore the effects of pandemic even though many precautions were reversed recently. Moreover, the exact time for the exact reversal of pandemic is not known.

Besides measures taken until now, additional national measures for better integration and more mental support for Syrian HD patients are needed.

## Acknowledgments

Authors thank to Dr Alp Ikizler for his help during proofreading.

## Author Contributions

**Conceptualization:** Mustafa Sevinc, Nuri Baris Hasbal, Tamer Sakaci, Elbis Ahbap, Emrah Erkan Mazi, Efruz Pirdogan.

**Data curation:** Mustafa Sevinc, Nuri Baris Hasbal, Tamer Sakaci, Elbis Ahbap, Mustafa Orta-boz, Emrah Erkan Mazi, Efruz Pirdogan.

**Formal analysis:** Nuri Baris Hasbal, Tamer Sakaci, Emrah Erkan Mazi, Efruz Pirdogan.

**Investigation:** Tamer Sakaci, Elbis Ahbap, Efruz Pirdogan.

**Methodology:** Mustafa Sevinc, Nuri Baris Hasbal, Tamer Sakaci, Efruz Pirdogan.

**Project administration:** Tamer Sakaci.

**Software:** Nuri Baris Hasbal, Tamer Sakaci.

**Supervision:** Tamer Sakaci, Taner Basturk, Elbis Ahbap, Jonathan Ling, Abdulkadir Unsal.

**Validation:** Tamer Sakaci, Mustafa Ortaboz.

**Visualization:** Tamer Sakaci.

**Writing – original draft:** Mustafa Sevinc.

**Writing – review & editing:** Mustafa Sevinc, Nuri Baris Hasbal, Tamer Sakaci, Taner Basturk, Jonathan Ling, Abdulkadir Unsal.

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
