## [Decision Letter · Decision Letter 0]

25 Nov 2020

PONE-D-20-26043

Effect of COvid-19 on mental health in Syrian and Turkish maintenance HemoDialysis patients: COST-HD study

PLOS ONE

Dear Dr. Sakaci,

Thank you for submitting your manuscript to PLOS ONE. After careful consideration, we feel that it has merit but does not fully meet PLOS ONE’s publication criteria as it currently stands. Therefore, we invite you to submit a revised version of the manuscript that addresses the points raised during the review process.

We apologize for the delay in our response. We have had some difficulties with the reviewers of this work and that is the reason why I have reviewed the work mainly on my own. You will find my comments below.

We look forward to receiving your revised manuscript.

Kind regards,

Jose A. Muñoz-Moreno, Ph.D.

Academic Editor

PLOS ONE

Journal Requirements:

2.) Please amend your current ethics statement to address the following concerns: Please explain why written consent was not obtained, how you recorded/documented participant consent, and if the ethics committees/IRBs approved this consent procedure.

3) In your ethics statement in the Methods section and in the online submission form, please provide additional information about the retrospective demographic data used in your study. Specifically, please ensure that you have discussed whether all data were fully anonymized before you accessed them and/or whether the IRB or ethics committee waived the requirement for informed consent. If patients provided informed written consent to have data from their medical records used in research, please include this information.

4.) Please include the date(s) on which you accessed the databases or records to obtain the retrospective demographic data used in your study.

5.) We note that you have stated that you will provide repository information for your data at acceptance. Should your manuscript be accepted for publication, we will hold it until you provide the relevant accession numbers or DOIs necessary to access your data. If you wish to make changes to your Data Availability statement, please describe these changes in your cover letter and we will update your Data Availability statement to reflect the information you provide.

**Comments to Authors**

1. Is the manuscript technically sound, and do the data support the conclusions?

Reviewer #1: Partly

2. Has the statistical analysis been performed appropriately and rigorously? 

Reviewer #1: Yes

3. Have the authors made all data underlying the findings in their manuscript fully available?

Reviewer #1: Yes

4. Is the manuscript presented in an intelligible fashion and written in standard English?

Reviewer #1: Yes

5. Review Comments to the Author

Reviewer #1: - The tittle should not mention "mental health", since the study is only investigating depressive symptoms and many other dimensions of mental health are not addressed.

- I would also recommend including in the tittle "refugees", since this is a key point of the study.

- It is mentioned that pneumonia of unknown cause was detected on 31 December 2019, but it was reported on December 30th: https://promedmail.org/promed-post/?id=6864153

- Importantly, all the manuscript refers to "depression". But this work did not investigate depression rigorously, but a screening of depressive symptoms. Depression should be thoroughly diagnosed by a clinical interview, or alternatively, in the case of research setting, with more complete methodology. BDI is merely a method to measure depressive symptoms, with relevant limitations in fact. Authors should be really cautious on this point.

- In Abstract, it is reported that demographic and clinical data were collected from patients' files; what type of files? This feature should be clarified. Adapt also in the manuscript (Methods).

- Which version of BDI was used? An appropriate reference should be cited the first time it is mentioned in text.

- In Abstract, when BDI means are reported, the numbers in parenthesis are not defined.

- In Abstract, the last sentence in Results should clarify whether those factors were found in the total sample or only in one of the groups.

- In Abstract, Conclusions, it is stated that HD patients were affected by the pandemic, but this work did not investigate any aspect about impact of COVID-19 specifically. The study was cross-sectional, with no prospective data, therefore authors were not allowed to conclude any information on the affectation by the pandemic. If there is any reference of some work previously published in that regard, it could be provided.

- Introduction, line 73, it is stated that depression can be diagnosed by questionnaires. Again, depression should be carefully diagnosed with the help of questionaries, alongside a proper psychiatric interview. In fact, the citation provided in the text is referring to screening for depression very clearly.

- Introduction, line 75, "Syrian civil" is missing a word.

- In the Introduction, allusions to impact of the pandemic are made, but, again, the study essentially offers a picture of the depressive symptoms.

- Methods, line 82, the date should be adapted to English format. Importantly, all dates provided in the Abstract and manuscript should be adapted as well.

- Methods, line 96, "was accepted as suspicion of depression".

- Results, line 121, "due to following".

- Results, line 122, "COVID-19".

- Results, line 131, sentence should be rephrased.

- Results, line 133, "depression" should be changed to "depressive symptoms." Also in in the Abstract and all along the manuscript.

- 50.7% of the Turkish cohort were women, 18.2% in the Syrian cohort (p=0.014). And depressive symptoms were more prevalent in the Syrian population (p=0.03). However, when BDI outcomes were stratified according to a score of 14, the proportion of female patients did not differ. This result should be confirmed.

- In Results, firstly, a description of the cohorts is offered; later, both groups are joined to study the factors related to the existence of suspicion of depression (BDI score of 14); and finally, again, groups are separated to offer the difference in BDI means according to groups. The third part should be provided after the first one, and factors related to suspicion of depression should be the last one. In fact, that is the order followed in the Abstract.

- Discussion, line 153, "male, although."

- Discussion, line 153, 54% is with no decimal, and 51.6% include it.

- The paragraph concerning the description of BDI and its potential usefulness is really scarce. Other advantages should be added, and, importantly, its main limitations; for example, that it is widely biased by organic symptoms, or that the original BDI version has been updated by BDI-II years ago.

- Outcomes on BDI could be presented as 2 subscales (cognitive-affective and somatic-performance). Because only one questionnaire has been applied in the study, incorporating this information to the manuscript could definitively offer more strength to the work.

- Proportion of women was significantly higher in the Syrian group compared to the Turkish group. Depressive symptoms were also higher in the same group. This could be perfectly attributed then to commonly higher rates of depressive status in the general population, or in refugee women as well. This point should be properly justified and argued in Discussion.

- In Discussion, alongside the general recommendation to provide mental support to Syrian refugees, other specific suggestions could be made. In my opinion, this part of the Discussion is truly important, since authors should use their knowledge and experience to state clearly specific interventions that could be delivered and implemented.

- Table 1: Kt/V mean for all patients, there is a mistake in the decimal.

- Table 1: one decimal, two, or three for data shown? This is also important for the Abstract and along the manuscript.

- Figure 1: format mistakes should be corrected.

- Figure 1, title: "Details of the patient cohort."

- Figure 2: p value could be included in the graph.

- Finally, and very importantly, English grammar and style should be revised for all the text.

6. PLOS authors have the option to publish the peer review history of their article (what does this mean?). If published, this will include your full peer review and any attached files.

Reviewer #1: No

---

## [Author Response · Author response to Decision Letter 0]

8 Dec 2020

Response to reviewers

Dear reviewer,

Thank you very much for your review and contributions. 

Your contributions and our answers can be seen below.

Kind regards

Tamer Sakaci

Journal requirements

Answer:

 Title page and manuscript have been updated regarding Journal’s rules. 

2.) Please amend your current ethics statement to address the following concerns: Please explain why written consent was not obtained, how you recorded/documented participant consent, and if the ethics committees/IRBs approved this consent procedure.

Answer:

We are sorry for forgetting to add this detail to the manuscript. All participants had given written consent when they had filled the BDI form. This detail has been added to the first paragraph of method section (lines 67-69 at manuscript file) as “Participants signed written consent forms both to participate in the study and to have data from their medical records used in research.” 

3) In your ethics statement in the Methods section and in the online submission form, please provide additional information about the retrospective demographic data used in your study. Specifically, please ensure that you have discussed whether all data were fully anonymized before you accessed them and/or whether the IRB or ethics committee waived the requirement for informed consent. If patients provided informed written consent to have data from their medical records used in research, please include this information.

Answer:

We are sorry for skipping to add these details. Data were anonymized before our access. Consent form was containing details to have data from patients’ medical records. The sentence regarding these items has been added to the first paragraph of methods section (line 67 at manuscript file) as follows: “All data were fully anonymized before access. Participants signed written consent forms both to participate in the study and to have data from their medical records used in research. “ 

4.) Please include the date(s) on which you accessed the databases or records to obtain the retrospective demographic data used in your study.

Answer

The data was accessed at the time of BDI questionnaire. It has been added to the end of second paragraph of methods section (lines 75,76 at manuscript file) as “Past medical records were accessed at the same day patients completed the BDI questionnaire.” 

5.) We note that you have stated that you will provide repository information for your data at acceptance. Should your manuscript be accepted for publication, we will hold it until you provide the relevant accession numbers or DOIs necessary to access your data. If you wish to make changes to your Data Availability statement, please describe these changes in your cover letter and we will update your Data Availability statement to reflect the information you provide.

 Answer: 

We have uploaded our date via Dryad system by doi:10.5061/dryad.wwpzgmshv. The data will be open upon acceptance of the article. The link to access data is https://datadryad.org/stash/share/unPym5dTZ-c9G-A1p6cbGxD_OEsTUTdPlzYhJQXbO5M

Comments to Authors

5. Review Comments to the Author

Reviewer #1: - The tittle should not mention "mental health", since the study is only investigating depressive symptoms and many other dimensions of mental health are not addressed.

Answer:

The title is updated as “Frequency of depressive symptoms in Syrian refugees and Turkish maintenance hemodialysis patients during COVID-19 pandemic”

- I would also recommend including in the tittle "refugees", since this is a key point of the study.

Answer:

The title is updated as “Frequency of depressive symptoms in Syrian refugees and Turkish maintenance hemodialysis patients during COVID-19 pandemic”

- It is mentioned that pneumonia of unknown cause was detected on 31 December 2019, but it was reported on December 30th: https://promedmail.org/promed-post/?id=6864153

Answer:

We are sorry this mistake. It was corrected as 30 December 2019 at abstract and introduction section. 

- Importantly, all the manuscript refers to "depression". But this work did not investigate depression rigorously, but a screening of depressive symptoms. Depression should be thoroughly diagnosed by a clinical interview, or alternatively, in the case of research setting, with more complete methodology. BDI is merely a method to measure depressive symptoms, with relevant limitations in fact. Authors should be really cautious on this point.

Answer: 

Thank you for this advice. The abstract and manuscript is changed regarding your suggestion. Depressive symptoms and suspicion of depression are the terms used instead of depression. 

- In Abstract, it is reported that demographic and clinical data were collected from patients' files; what type of files? This feature should be clarified. Adapt also in the manuscript (Methods).

Answer: 

Details of these files have been added to abstract (lines 25-27 at manuscript file) as follows: “Demographic and clinical data were collected retrospectively from patients’ files containing details about past medical history, demographic variables and laboratory values.” 

Definitions and details of the files has been added to the methods section (lines 77-79 at manuscript file) as well: “Demographic data were collected retrospectively from patients’ files. These files contain the demographic details of patients filled at admission to our dialysis center and patients’ past and current laboratory values.”

- Which version of BDI was used? An appropriate reference should be cited the first time it is mentioned in text.

Answer:

The original version of BDI published at 1961 was used. It is referred as well (reference number 11). The text has been updated (lines 88,89 at manuscript file) as follows: “The validated Turkish and Arabic forms of BDI version I were filled by patients themselves [9-11].”

- In Abstract, when BDI means are reported, the numbers in parenthesis are not defined.

Answer:

We are sorry for mistake. Abstract has been updated (lines 32, 33 at manuscript file) as “The median BDI (interquartile range) score for Turkish and Syrian patients were 12 (7-23) and 19.5 (12.7-25.2), respectively (p=0.03).”

- In Abstract, the last sentence in Results should clarify whether those factors were found in the total sample or only in one of the groups.

Answer:

It was found in the total sample. This sentence (lines 35-37 at manuscript file) has been changed as “Regarding all patients, phosphorus level, Kt/V, and nationality were significantly different between patients with and without depression (p=0.023, 0.039, 0.013, respectively).”

- In Abstract, Conclusions, it is stated that HD patients were affected by the pandemic, but this work did not investigate any aspect about impact of COVID-19 specifically. The study was cross-sectional, with no prospective data, therefore authors were not allowed to conclude any information on the affectation by the pandemic. If there is any reference of some work previously published in that regard, it could be provided.

Answer:

Thank you for this missing point. We agree you. The first part of the first sentence of conclusion section has been deleted (lines 38-40 at manuscript file) as follows: “Syrian patients had higher BDI scores and more depressive symptoms than Turkish patients. Additional national measures for better integration and more mental support to Syrian HD patients are needed. ” 

- Introduction, line 73, it is stated that depression can be diagnosed by questionnaires. Again, depression should be carefully diagnosed with the help of questionaries, alongside a proper psychiatric interview. In fact, the citation provided in the text is referring to screening for depression very clearly.

Answer:

Thank you very much for this caution. We agree you. The sentence (lines 55, 56 at manuscript file) changed as “It can be screened by a few types of self-questionnaires validated in this population such as Beck depression inventory (BDI)”.

- Introduction, line 75, "Syrian civil" is missing a word.

Answer: 

It has been completed (line 58 at manuscript file) as “Syrian civil war has been started on 11 March 2011.”

- In the Introduction, allusions to impact of the pandemic are made, but, again, the study essentially offers a picture of the depressive symptoms.

Answer:

The make this effect less, the sentence at the introduction section “We do not know if pandemic affected Syrian patients’ mental health different than Turkish ones” has been changed (lines 60-62 at manuscript file) as “We do not know if there is a difference in depressive symptom frequency between Syrian refugee HD patients and Turkish ones or not”. 

- Methods, line 82, the date should be adapted to English format. Importantly, all dates provided in the Abstract and manuscript should be adapted as well.

Answer:

We agree you. That date and the others are all adapted. 

- Methods, line 96, "was accepted as suspicion of depression". 

Answer:

We agree you. The sentence (line 87 at manuscript file) is corrected as your suggestion. 

- Results, line 121, "due to following".

Answer:

We are sorry for that. It has been corrected. 

- Results, line 122, "COVID-19".

Answer:

We are sorry for that. It has been corrected. 

- Results, line 131, sentence should be rephrased.

Answer:

It has been rephrased (line 127 at manuscript file) as “Marriage rate was 49.3% in Turkish and 81.8% in Syrian patients”. 

- Results, line 133, "depression" should be changed to "depressive symptoms." Also in in the Abstract and all along the manuscript.

Answer:

All text is updated regarding your advice. Thank you for this recommendation. 

- 50.7% of the Turkish cohort were women, 18.2% in the Syrian cohort (p=0.014). And depressive symptoms were more prevalent in the Syrian population (p=0.03). However, when BDI outcomes were stratified according to a score of 14, the proportion of female patients did not differ. This result should be confirmed.

Answer:

The tests and results are confirmed again. The results seem correct. The confusing fact might be that the when BDI scores were stratified, nationalities were not taken into account. Grouping variable in this comparison is BDI stratification, not nationality. 

More analysis has been done to make it clear as follows: 

Frequency of depressive symptoms in Syrian female patients was similar to Turkish female patients (p=0.05). It might be due to limited number of participants because p value is close to the significance level. It was significantly different between nationalities in male patients (p=0.034). 

- In Results, firstly, a description of the cohorts is offered; later, both groups are joined to study the factors related to the existence of suspicion of depression (BDI score of 14); and finally, again, groups are separated to offer the difference in BDI means according to groups. The third part should be provided after the first one, and factors related to suspicion of depression should be the last one. In fact, that is the order followed in the Abstract. 

Answer: 

The flow of results has been changed regarding your recommendation. 

- Discussion, line 153, "male, although."

Answer: 

It is corrected. Thank you. 

- Discussion, line 153, 54% is with no decimal, and 51.6% include it.

Answer: 

The sentence (lines 156-158 at manuscript file) has been changed as “In our cohort, most of the Syrian patients (81.8%) were male, although 54.0% of all Syrian refugees [14] and 51.6 % of Syrian hemodialysis patients [5] in Turkey were male.”

- The paragraph concerning the description of BDI and its potential usefulness is really scarce. Other advantages should be added, and, importantly, its main limitations; for example, that it is widely biased by organic symptoms, or that the original BDI version has been updated by BDI-II years ago.

Answer: 

Thank you for this contribution. The paragraph (lines 159-170 a manuscript file) has been updated as follows: 

“Screening frequency of depressive symptoms by the BDI score without a clinical interview is an established strategy in ESRD patients [3, 7, 8]. BDI-I questionnaire was used in this study because of some advantages. It was a practical, easy, reliable and valid self-report system that has been used since 1961 [11, 15]. It could be completed by patients themselves which would minimize close and long contacts with patients during the pandemic and prevent potential COVID-19 transmission. Moreover, to the best of our knowledge, it is the only self-report screening measure that has been validated in both Turkish and Arabic languages. Besides these advantages, BDI-I form has some limitations. It has been focused more on cognitive and affective symptoms than somatic functions. Furthermore, it could be biased by organic symptoms. It was updated as BDI-II form at 1996 in response to American Psychiatric Association’s publication of DSM-IV criteria for major depressive disorder [16]. Despite this updated version, BDI-I is still widely used in different patient populations [17]. “

- Outcomes on BDI could be presented as 2 subscales (cognitive-affective and somatic-performance). Because only one questionnaire has been applied in the study, incorporating this information to the manuscript could definitively offer more strength to the work.

Answer:

We added data regarding BDI subscales to the relevant sections as follows: 

Method section: 

These two sentences (lines 97, 98 at manuscript file) were added at the end of methods section: “BDI score was also divided into cognitive-affective and somatic-performance subscales [12]. These were compared to nationality as well.” 

Results section: 

The new findings are added (lines 129-132 at manuscript file) as follows: The median somatic-performance subscale scores for Turkish and Syrian patients were 2 (1-5) and 3.5 (2-6.2), respectively (p=0.02). Affective-cognitive subscale median score for Turkish patients was 9 (6-18) whereas it was 14 (10-18) for Syrian patients (p=0.05).

Discussion section: 

Some additions and rephrasing were done as follows (lines 194-198 at manuscript file): “However, providing equal physical and financial conditions only at peridialysis environment seemed to be insufficient for equal score expectation because we found that total BDI score, somatic-performance subscale score and affective-cognitive subscale score were all higher in Syrian patients than Turkish ones. As a result, it seems that we need to provide many other facilities to talk about equality between the two groups.”

- Proportion of women was significantly higher in the Syrian group compared to the Turkish group. Depressive symptoms were also higher in the same group. This could be perfectly attributed then to commonly higher rates of depressive status in the general population, or in refugee women as well. This point should be properly justified and argued in Discussion.

Thank you for this comment. 

The Syrian female patients was 18.2% of their cohort (n=4) and the Turkish female patients was 52% of their cohort (n=39). 

Regarding all patients, depressive symptom frequency was similar between gender (p=0.482). Regarding only females, all Syrian females and 48.7% of Turkish females had depressive symptoms (p=0.05). statistical significance may not be reached due to limited number of patients. Regarding only Syrian patients, median BDI score for males was 17 (11.7-23.5). It was 23.5 (19.5-30.5) (p=0.141). Comparison of gender in Syrian patients against BDI score ≤14 or >14 was not significant (p=0.176). As a result, even though all Syrian females had depressive symptoms, its frequency was similar to Syrian males. 

- In Discussion, alongside the general recommendation to provide mental support to Syrian refugees, other specific suggestions could be made. In my opinion, this part of the Discussion is truly important, since authors should use their knowledge and experience to state clearly specific interventions that could be delivered and implemented.

Answer: 

Thank you for this contribution. The related paragraph (lines 189-205 at manuscript file) is changed as follows: 

“Yilmaz et al have shown that Syrian maintenance HD patients were less compliant with their HD schedules [35]. Their per capita income was lower than Turkish patients, although their employment rate was higher. The number of Syrian household members was higher than Turkish patients but hot water accessibility was lower for the Syrian group. Similar BDI scores might be expected between Turkish and Syrian patients if the conditions were the same for both groups. However, providing equal physical and financial conditions only at peridialysis environment seemed to be insufficient for equal score expectation because we found that total BDI score, somatic-performance subscale score and affective-cognitive subscale score were all higher in Syrian patients than Turkish ones. As a result, it seems that we need to provide many other facilities to talk about equality between the two groups. These might be steps to improve language barrier between adult Syrian patients and Turkish service providers at all facilities including hospitals and other institutions. This could be with two methods: enabling Syrians to learn Turkish and maintaining more translation support at every institution or integrating native Syrian employees to these institutions like hospitals. Language support would help integration with the Turkish population. Most importantly, they should have been accepted and behaved like as others in Turkey, not as refugees. Integrating 4 million people into an economically hard stressed country in the midst of a pandemic is not easy but it seems as a necessity “.

- Table 1: Kt/V mean for all patients, there is a mistake in the decimal.

Answer: 

I am sorry for this. It is corrected as “1.7±.0.2”. 

- Table 1: one decimal, two, or three for data shown? This is also important for the Abstract and along the manuscript.

Answer: 

Abstract, all parts of the manuscript, tables are updated as one decimal format.

- Figure 1: format mistakes should be corrected.

Answer:

They are corrected. 

- Figure 1, title: "Details of the patient cohort."

Answer:

It is corrected.

- Figure 2: p value could be included in the graph.

Answer:

It is added. 

- Finally, and very importantly, English grammar and style should be revised for all the text.

Answer:

Thank you for this comment. It has been rechecked by us and a native speaker, Jonathan Ling. Jonathan has reviewed the manuscript and revised it regarding your suggestions. Due to these reasons, the authors of this manuscript think that he deserves to be an author is this article. Could you please add Jonathan Ling as a co-author of this article? ________________________________________

---

## [Editor Report · Decision Letter 1]

9 Dec 2020

Frequency of depressive symptoms in Syrian refugees and Turkish maintenance hemodialysis patients during COVID-19 pandemic

PONE-D-20-26043R1

Dear Dr. Sakaci,

We’re pleased to inform you that your manuscript has been judged scientifically suitable for publication and will be formally accepted for publication once it meets all outstanding technical requirements.

Kind regards,

Jose A. Muñoz-Moreno, Ph.D.

Academic Editor

PLOS ONE

---

## [Editor Report · Acceptance letter]

11 Dec 2020

PONE-D-20-26043R1 

Frequency of depressive symptoms in Syrian refugees and Turkish maintenance hemodialysis patients during COVID-19 pandemic 

Dear Dr. Sakaci:

I'm pleased to inform you that your manuscript has been deemed suitable for publication in PLOS ONE. Congratulations! Your manuscript is now with our production department. 

Kind regards, 

on behalf of

Dr. Jose A. Muñoz-Moreno 

Academic Editor

PLOS ONE